# Mortality Patterns in a Commercial Wean-To Finish Swine Production System

**DOI:** 10.3390/vetsci6020049

**Published:** 2019-05-29

**Authors:** Samantha Mehling, Alexandra Henao-Diaz, Jeremy Maurer, Ed Kluber, Rachel Stika, Christopher Rademacher, Jeff Zimmerman, Luis Giménez-Lirola, Chong Wang, Derald J. Holtkamp, Rodger Main, Locke Karriker, Daniel C L Linhares, Mary Breuer, Christa Goodell, David Baum

**Affiliations:** 1Department of Veterinary Diagnostic and Production Animal Medicine, College of Veterinary Medicine, Iowa State University, Ames, IA 50011, USA; smehling@iastate.edu (S.M.); yulyh@iastate.edu (A.H.-D.); rstika@iastate.edu (R.S.); cjrdvm@iastate.edu (C.R.); jjzimm@iastate.edu (J.Z.); luisggl@iastate.edu (L.G.-L.); chwang@iastate.edu (C.W.); holtkamp@iastate.edu (D.J.H.); rmain@iastate.edu (R.M.); karriker@iastate.edu (L.K.); linhares@iastate.edu (D.C.L.L.); mebreuer@iastate.edu (M.B.); 2Smithfield Foods, Milford, UT 84751, USA; jmaurer@smithfield.com (J.M.); ed.kluber@gmail.com (E.K.); 3IDEXX, Westbrook, ME 04092, USA; christa.goodell@boehringer-ingelheim.com

**Keywords:** epidemiology, wean-to-finish, economic, impact, mortality, pattern, stocking density

## Abstract

Modern commercial pig production is a complex process that requires successful producers to understand and resolve factors associated with perturbations in production. One important perturbation is inventory loss due to mortality. In this study, data on 60 lots of approximately 2000 weaned pigs (n = 115,213) from one commercial production system were collected through the wean-to-finish (WTF) cycle with the objective of establishing patterns of mortality, estimating differences in profit/loss among patterns of mortality, and identifying production practices associated with mortality patterns. Information provided by the production system included the number of pigs in each lot at the time of placement (beginning inventory), weaning weight, barn dimensions, number of dead pigs (NDP) daily, capacity placed (proportion pigs actually placed versus what had been planned to be placed) and average weight sold. Analysis of NDP revealed three mortality patterns (clusters I, II, III) composed of 6, 40, and 14 lots, respectively, that differed in the temporal onset and/or level of mortality. Average daily gain (ADG) and feed conversion ratio (FCR) were calculated by growth phase for each cluster. An economic model showed profit differences among clusters due to poor biological performance by clusters I and III in the late finishing phase. Cluster II (n = 40) had fewer dead pigs and the highest profit compared to clusters I (n = 6) and III (n = 14). Area per pig (stocking density) was the only factor associated with the differences in mortality patterns. Routine monitoring and the analysis of mortality patterns for associations with production and management factors can help swine producers improve biological performance and improve profit.

## 1. Introduction

Modern pig production is a complex process that results in pounds of pork generated over time. Optimum performance requires production systems to understand factors associated with perturbations in production during the course of the post-weaning period, including the inventory loss due to pig deaths. Previous studies of post-weaning pig deaths described infectious causes [1,2], pig mortality patterns [1,3], and risk factors [2,4,5,6]. One study identified mortality reduction as an animal welfare issue and highlighted the need to identify risk factors [4], sources of variation at the pig and barn level [4,5,6], or associated pathogens via surveillance [2]. Each paper advised pork producers to investigate their own production systems for factors associated with increased mortality. This study sought to determine if distinct mortality patterns existed within a commercial wean-to-finish (WTF) system, the profit/loss impact those patterns might have, and what non-pathogen factors might be associated with mortality patterns. Such an outcome will encourage swine health management teams to account for those non-pathogen factors while seeking to improve swine health.

## 2. Materials and Methods 

### 2.1. Experimental Design

This field study was approved by the Iowa State University Office for Responsible Research Institutional Animal Care and Use Committee (IACUC) (IACUC log # 2-15-7959-S) and supervised by Smithfield Foods’ Health Assurance and Welfare System. 

Dead pigs were recorded daily throughout the WTF period of 60 lots of pigs, each lot consisting of ~2000 pigs. Additional information included barn and pen dimensions, number of pigs per pen, weaning weight, number of dead pigs (NDP) daily, capacity placed (proportion pigs actually placed versus what had been planned to be placed), and average weight sold.

### 2.2. Animals, Animal Care, and Management

The production facility managers and senior veterinary staff provided animal care. The study was performed in an 80,000-sow commercial production system that provided weaned pigs to contract nurseries or WTF facilities in Midwest and West USA. Porcine reproductive and respiratory disease virus (PRRSV), influenza A virus (IAV), porcine epidemic diarrhea virus (PEDV), porcine circovirus type 2 (PCV2), *Mycoplasma hyopneumoniae* (MHP) and *Actinobacillus pleuropneumoniae* (APP) were endemic to the system. 

Sixty lots of pigs monitored in the study were weaned from six 5000-sow breeding-gestating-farrowing (BGF) farms between June 10 2015 and December 30 2015. Within a seven-day period, commingled weaned piglets from these six BGF farms created a ‘lot’, i.e., ~2000 pigs weaned at approximately 25 days of age and housed within one barn in one of eight WTF sites. Six of these sites were located in Area 1 and identified sequentially as 1–6. The other two sites were located approximately three miles away, in area 2 and identified as 7 and 8 (Figure 1). Forty-eight lots were housed simultaneously among the eight sites. The final 12 lots of pigs were housed at sites 4 and 6. There were six lots at each site, because the previous lots in these sites had been sold.

All buildings were fully-slatted with hairpin gutters. Pigs were fed the same diet formulations throughout their WTF period. Diets were ad libitum via an automated feed delivery system that delivered feed from bulk feed bins located at the end of each barn into overhead distribution lines and into shared fence-line feeders. Free access to water was available in each pen by swinging waterers. Ventilation fans and curtain siding spanning the north and south sides of each barn provided temperature control. Site managers flushed manure from manure pits weekly and refilled the pits with water. Site managers daily removed dead pigs from the barns, placed them into on-site containers, and recorded the number of dead pigs (NDP) for each lot. Calculations of average weaning weight were made at the time of weaning. Average sale weight was calculated for each lot after the sale of all pigs from the lot.

### 2.3. Site Configuration

Each WTF site consisted of six barns (Figure 2) oriented east-to-west and connected by a central north-to-south facing hallway. Three equally-spaced barns were located on each side of the central hallway (west 1–3 and east 1–3). Each barn at site 7 contained 38 pens (6.1 m × 5.79 m), with pens 1, 19, 20, and 38 designated as treatment pens for sick animals. Each barn at sites 1–6, and 8 contained 46 pens (4.98 m × 5.79 m) with pens 1, 23, 24, and 46 designated as treatment pens. Treatment pen dimensions were one-half the area of the housing pens. Figure 3 shows the numbering and configuration of individual pens for each barn model. Table 1 provides additional barn dimensions for each site.

### 2.4. Phase Cost Calculations per Cluster

In order to estimate between-cluster cost differences, the WTF period was split into four phases based upon days per phase. (R. Stika, 2018, personal communication): (1) placement (days 0–28), (2) nursery (days 29–56), (3) early finishing (days 56–112), and (4) late finishing (day 112–finish). Weaned pig starting weight, average daily gain, and average daily feed intake assumptions determined average daily gain and, therefore, pig weight at the end of each diet throughout the cluster (Table 2). Seven diets were used in this model, each with successively decreasing costs per kilogram. The change to the next sequential diet depended on the number of days of the previous diet. The estimated final market weight was compared to the average of the market weights reported by the farm for each cluster. These were made to match by multiplying the expected kg gained by a factor for each diet fed during the time when mortality appeared to increase. The factors were applied relative to the amount of mortality while fed the respective diets. The feed costs and final average weights of each cluster were thus calculated and used in the profit/loss calculations for each cluster.

### 2.5. Profit/loss Calculations per Cluster

An economic model for calculation of the profit/loss (model) for each cluster by phase was adapted from a spreadsheet prototype (model created by D. Holtkamp, 2018, personal communication) and reported as profit/loss per 1000 head. Except for the calculated feed costs and mortality per phase, the same variable and fixed costs were assigned to each cluster. The model calculated feed costs from ingredient costs for a corn-soy diet, which in turn drove the calculation of feed cost per kg. The calculation of mortality for each phase was equal to the number of pigs that died during the phase divided by the phase’s beginning pig inventory.

### 2.6. Statistical Analysis

The general linear model of SAS^®^ (Cary, NC, USA) was used to compare clusters’ starting weights, head placed, capacity placed, area (m^2^ per pig) and mortality. 

## 3. Results

This section may be divided by subheadings. It should provide a concise and precise description of the experimental results, their interpretation as well as the experimental conclusions that can be drawn.

### 3.1. Creation of Mortality Clusters

A time series plot of each of the 60 lots’ mortality by week on feed was created and lots with observed similarities in mortality patterns were assigned to a cluster. Thus, a cluster included all lots in which increased mortality appeared to begin during approximately the same week(s) on feed (Figure 3). Based on this procedure, cluster I contained six lots with higher mortality between weeks 14 and 20; cluster III consisted of 14 lots with higher NDP between weeks 18 and 23; and cluster II was composed of 40 lots with relatively low NDP for the entire time on feed. These three patterns were used as the output variables for this study: increased NDP between weeks 13 and 17 (cluster I, n = 6 lots), low NDP throughout all weeks on feed (cluster II, n = 40 lots), and increased NDP after week 17 on feed (cluster III, n = 14 lots).

Cluster I consisted of six lots housed at one site (Site 8). Cluster II consisted of 40 lots from Site 2–Site 6, six lots each, and five lots each from Site 1 and Site 7. Cluster III consisted of 14 lots from Site 4 and Site 6, six lots each, and one lot each from Site 1 and Site 7.

Sites 4 and 6 each housed two complete WTF cycles, thus 12 lots per site for the duration of this study. Each site’s first WTF cycle lots’ mortality were part of cluster III and each site’s second WTF cycle lots’ mortality were part of cluster II.

### 3.2. Production and Cost Analysis

Average mortality for each cluster by phase is reported in Table 2. Cluster II began with higher mortality in the Placement phase compared with clusters I and III and remained similar to cluster III in the Nursery and Early Finishing phases. Cluster I mortality begins to increase in the Nursery phase, remains similar in the Early Finishing phase and ends with an average of 23.87% in the Finishing phase. Cluster III mortality is greater than cluster II but less than cluster I in the Finishing Diet costs and performance assumptions used to model feed costs per pig are in Table 3. This enabled accounting of diets that spanned more than one phase of growth (Table 4). Table 4 includes the market weight expected from the input assumptions of Table 3.

#### Model Adjustments

Average days on feed, average weight at placement, and average daily gain determined “average weight at market” for the end of each of the model’s phases: placement, nursery and early finishing. Each cluster’s final sale weight was determined after the late finishing phase. Adjustments to the average daily gain adjusted the model’s expected cumulative weight (ExpCumWt, kg) to match the average market weight reported from the farm’s data for each lot within each cluster. Then, a comparison was made of a cluster’s modeled final market weight with the average of market weights reported by the farm for each lot in that cluster. The modelled average market weight was made to match the reported market weight, by multiplying ADG factors by the expected ADG. Table 5 is an example of how ADG adjustments were made to match the actual market weight for cluster I with a wean weight of 7.14 kg. Table 6 is an example of the factors’ impact on expected end weight, days on feed and feed costs.

Each cluster’s phase feed conversion ratio (FCR) were then calculated (Table 7). Cluster II’s average daily gain was numerically greater than clusters I and III throughout the mode. The FCR of cluster II was numerically lower than that of cluster I and III; thus, cluster II pigs’ modelled performance was more efficient in converting feed to body weight. The economic model used these values to calculate each cluster’s phase feed costs (FC, Table 8). Cluster II had the highest total FC per pig compared to the other two clusters. Cluster II had more surviving pigs to eat feed.

The modelled profit for each cluster appears in Table 9, wherein cluster II had lower expenses per pig and higher profit (less loss) and a higher average total pigs marketed than the other two clusters.

### 3.3. Cluster Management Data Characteristics

Average values for each cluster’s production management data are in Table 10. Cluster I had the highest starting weight, longest finishing period, and the lowest finishing weight per pig. Cluster I also had the highest percentage over capacity placed and lowest available square footage per pig. Cluster III was similar to cluster I with the same average starting weight, the same square footage per pig and the same capacity placed. Cluster III’s NDP was half (numerically) that of cluster I. Cluster II’s average starting weight was lower than clusters III and I, and its area per pig (m^2^/pig) was greater than the other two clusters. The NDP for cluster II was almost half that of cluster III and one fourth that of cluster I. 

## 4. Discussion

This study described distinct patterns of mortality by week on feed in this commercial WTF system. The patterns appeared empirically by the presence or absence of two phenomena: time and magnitude. Time, as observed by the week on feed when increased mortality began. Magnitude, as observed by the degree of increased mortality. Neither of these phenomena occurred in cluster II. Modelled profit for each cluster showed differences between lots (cluster II) with low mortality and those with high mortality (clusters I and III).

Increased pig stocking density has been described as a risk factor for mortality when multiple pig farm systems have been studied [4]. Jones [1], in an investigation of disease in a large commercial herd, hypothesized that the number of pigs populating a given house might have influenced the death rate and that the influence could be associated with the design and environment of the houses. However, Jones [1] could not determine possible associations of mortality with housing due to differences in the numbers of houses used and the movement of pigs from one house to another as space demanded. Dedecker et al. [7] investigated the impact of initial stocking rate on animal performance in a single-stage WTF production system under commercial conditions. This randomized complete block design study showed a linear increase in morbidity and mortality as group size increased from 22 to 27 to 32 pigs per 5.79 m × 3.05 m pen [7]. Categorical reasons for morbidity and mortality removal were proportionately similar across treatments [7]. This finding was important because of its animal welfare implications as well as a production output perspective [7]. 

We believe this is the first published observational study of production system data to describe pig mortality patterns in which pig-stocking density was associated with increased, and costly, mortality. Clusters I and III floor area per pig and mortality were statistically similar, even though their numerical values were 14.2% and 27.6%. This can be explained by variation in mortality and the different numbers of lots in each cluster, 6 and 14 lots, respectively. Floor area per pig was the only management factor associated with differences between cluster II’s low mortality and the high mortality of clusters I and III. This finding is important as it relates animal welfare to production output in a commercial setting. This study did not show a benefit to weaning heavier pigs as the weaning weight of cluster II was lower than either cluster I or cluster III and contrasts with the benefit of a heavier weaning weight and improved performance of pigs [8]. This contrast might be explained by the variations in pig stocking density observed in our study versus the constant pig stocking density of the study by Main et al. [8]. That is to say, each pen of Main et al., [8] was stocked with the same number of pigs (36 in Trial 1 and 34 in Trial 2). Together, these two studies may suggest that the benefit of heavier weaning weight may be lost if pigs are overstocked. Increased pig stocking density becomes an animal well-being matter as barn ventilation, water availability and feeder access are taxed to provide for animal comfort and growth. It may also suggest that when weaning weights are low, increasing floor space may reduce the mortality otherwise expected at lower floor space per pig.

The findings of this study are limited to the pig populations observed and the conditions in which they were raised. However, its methods are simple and can readily be adapted by any pig production system. The difficulty of adaptation lies in the need for dedicated and timely efforts in the study of a pig production system’s output and facility constraints. Weekly mortality data and previous years’ production levels and costs are readily available within a production system. Since pig housing area is readily determined, the impact of pig housing floor area (or other factors) can (and should) be analyzed post priori by the swine farm veterinary health team. Thus, data-driven recommendations can be used to inform management of factors associated with lost profit opportunities.

This study is meaningful because it shows that swine health management teams must view their important roles holistically. Holistic swine health management requires team members to understand and quantify environmental, management, and husbandry factors that affect pig performance, welfare, pathogens and profit. Cho and Kim’s [9] review paper on the effect of pig stocking density contains recommended space allowances for growing finishing pigs in Korea, Europe and the USA. These recommendations enable team members to estimate the costs of increased space allowances, the expected improvement in performance, and resultant increased pounds of pork sold. Such understanding and quantification enable them to teach these principles to everyone on the team, from those who get the daily work done, to those who are responsible to customers and shareholders.

## Figures and Tables

**Figure 1 vetsci-06-00049-f001:**
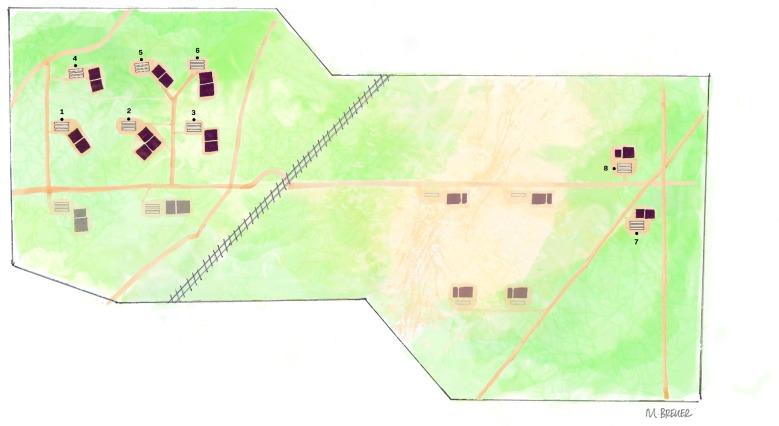
Aerial diagram showing wean-to-finish (WTF) sites of the production system studied. Area 1 (upper left) contains six WTF sites: 1, 2, 3, 4, 5 and 6. Area 2 (right) contains two WTF sites: 7, 8. Two of the six breeding-gestating-farrowing (BGF) sites that supplied the weaned pigs are located in the lower left below area 1.

**Figure 2 vetsci-06-00049-f002:**
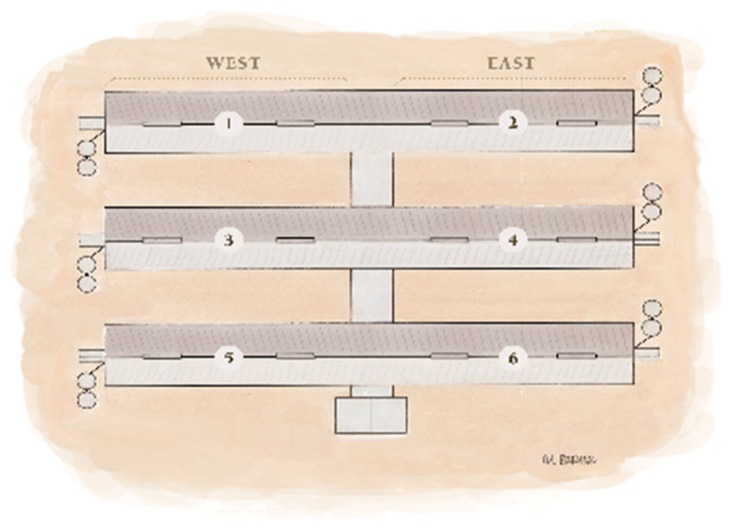
Configuration of a wean-to-finish site characteristic of this study: six barns per site, three on each side of a central hallway and numbered as shown. The central hallway connected all six barns. Feed bins were located at the end of each barn.

**Figure 3 vetsci-06-00049-f003:**
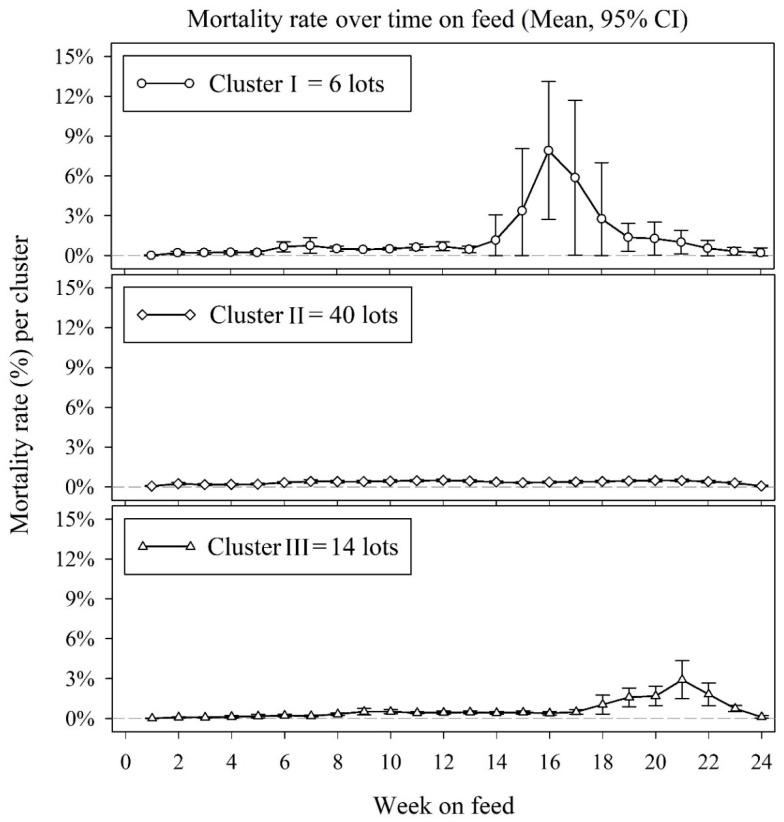
Cumulative mortality by week on feed. Clusters correspond to three mortality rate patterns: cluster I peak before week 17; cluster II constant low mortality over time; cluster III peak after week 17.

**Table 1 vetsci-06-00049-t001:** Square footage for each barn configuration, including area allotted per pen.

Site	Lots Per Barn	No. Pens	Pen Dimensions (m)	m^2^ per Pen	m^2^ per Barn
**1–6, 8**	1	46			1268.61
Housing pens		42	4.98 × 5.79	28.83	1211.04
Treatment pens		4	4.98 × 2.89	14.39	57.57
**7**	1	38			1271.40
Housing pens		34	6.1 × 5.79	35.32	1200.88
Treatment pens		4	6.1 × 2.89	17.63	70.52

**Table 2 vetsci-06-00049-t002:** Summary of mortality by phase expressed as a percentage of pigs placed at weaning. Phase is name of the period (weeks in phase) for four designations of pig growth used in the financial model.

Phase	Placement	Nursery	Early Finishing	Finishing
Weeks in Phase	1–4	5–8	9–12	13–24
Cluster I	0.02%	2.18%	2.28%	23.87%
Cluster II	0.69%	1.37%	1.81%	4.49%
Cluster III	0.29%	0.88%	1.86%	11.43%

**Table 3 vetsci-06-00049-t003:** Costs of diet and performance assumptions used to calculate feed costs per pig. Diets are designated by their phase and sequence (numeral) fed diet in the phase. N1 is the first nursery diet and F1 is the first finishing diet. The performance assumptions were expected kilograms average daily gain (Exp kg ADG), expected feed conversion rate (FCR), days on feed (days), expected kilograms average daily feed intake (ADFI), expected kilograms gained, expected kilograms cumulative weight (Exp kg Cum Wt) and diet cost per head. These values are for cluster I, where the average starting weight of its pigs was 7.14 kg.

Diet	Cost/kg	Exp kg ADG	Exp FCR	Days	Exp kg ADFI	Exp kg Gained	Exp kg Cum Wt	Diet Cost per Head
**N1**	$0.4134	0.36	1.45	9	0.53	3.27	10.40	$1.958
**N2**	$0.3307	0.43	1.65	14	0.71	6.03	16.43	$3.292
**N3**	$0.1929	0.57	1.90	14	1.08	7.94	24.37	$2.909
**F1**	$0.1653	0.70	2.10	21	1.48	14.76	39.14	$5.127
**F2**	$0.1450	0.79	2.48	30	1.95	23.61	62.75	$8.471
**F3**	$0.1361	0.87	3.05	31	2.64	26.86	89.60	$11.155
**F4**	$0.1207	0.79	3.73	74	2.96	30.96	148.34	$26.426

**Table 4 vetsci-06-00049-t004:** For each cluster, feed costs per pig were parsed into four phases based on diet assumptions in Table 2. These values are for cluster I.

Phase	Diet	Days	Exp kg ADFI	Exp kg Gain	Exp kg Cum Wt	FC	FC/Phase
**Placement**	N1	9	0.53	3.27	10.40	$1.96	$6.29
	N2	14	0.71	6.03	16.43	$3.29	
	N3	5	1.08	2.83	19.27	$1.04	
**Nursery**	N5	9	1.08	5.10	24.37	$1.87	$6.51
	F1	19	1.48	13.36	37.73	$4.64	
**Early Fin**	F1	2	1.48	1.41	39.14	$0.49	$17.60
	F2	30	1.95	23.61	62.75	$8.47	
	F3	24	2.64	20.79	83.54	$8.64	
**Late Fin**	F3	7	2.64	6.06	89.60	$2.23	$28.94
	F4	74	2.96	53.18	148.34	$26.43	

**Table 5 vetsci-06-00049-t005:** Cluster I’s feed and gain calculations and initial assumptions. The model ADG factors (ADG Factor) that adjusted the progression of pig weight through each phase of cluster I in order that expected market weight match actual average market weight (110.04 kg). Expected days on Diet F4 (Exp Days on Diet = 74) changed to match the actual average days on feed (92 days, see Table 5).

Diet	Cost/kg	Exp kg ADG	Exp FCR	Exp Days on Diet	Exp ADFI, kg	Exp kg Gained	ExpCumWt, kg	$Diet/Pig	ADG Factor
**N3**	$0.4134	0.36	1.45	9	0.53	3.27	10.40	$1.958	1.00
**N4**	$0.3307	0.43	1.65	14	0.71	6.03	16.43	$3.292	1.00
**N5**	$0.1929	0.57	1.90	14	1.08	7.94	24.37	$2.909	1.00
**F1**	$0.1653	0.69	2.10	21	1.48	14.03	38.84	$5.127	0.98
**F2**	$0.1450	0.77	2.48	30	1.95	22.43	61.98	$8.471	0.98
**F3**	$0.1361	0.43	3.05	31	2.64	13.43	75.41	$11.155	0.50
**F4**	$0.1207	0.39	3.73	39	2.96	15.17	90.58	$13.927	0.49

**Table 6 vetsci-06-00049-t006:** Cluster I feed and gain calculation after ADG factor adjustments. The result of increased days on diet F4 and ADG adjustments (Table 4) on feed costs and late finishing phase ending weight. Thus accounting for average days on feed and average market weight for cluster I, as reported by the farm.

Phase	Mortality	Diet	Days	Exp ADFI, kg	Exp kg Gained	Exp kg End wt	FC	FC/Phase
**Placement**	0.02%	N3	9	0.53	3.27	10.40	$1.96	$6.29
		N4	14	0.71	6.03	16.43	$3.29	
		N5	5	1.08	2.83	19.27	$1.04	
**Nursery**	2.18%	N5	9	1.08	5.10	24.37	$1.87	$6.51
		F1	19	1.48	12.69	37.06	$4.64	
**Early Fin**	2.28%	F1	2	1.48	1.34	38.40	$0.49	$17.60
		F2	30	1.95	22.43	60.83	$8.47	
		F3	24	2.64	10.40	71.22	$8.64	
**Late Fin**	23.87%	F3	7	2.64	3.03	74.26	$2.52	$35.37
		F4	92	2.96	35.78	110.04	$32.85	

**Table 7 vetsci-06-00049-t007:** The modelled average daily gain (ADG) and feed conversion ratio (FCR) for each cluster by phase are below. Cluster I and cluster III had the highest conversion ratio across all phases and the lowest ADG. Cluster II, the lowest FCR and the highest average daily gain.

	Placement		Nursery		Early Finishing		Late Finishing	
	ADG	FCR	ADG	FCR	ADG	FCR	ADG	FCR
**Cluster I**	0.73	2.17	1.1	2.69	1.37	3.58	1.34	4.81
**Cluster II**	0.94	1.69	1.42	2.09	1.77	2.78	1.74	3.69
**Cluster II**	0.82	1.94	1.24	2.39	1.54	3.19	1.51	4.26

**Table 8 vetsci-06-00049-t008:** Feed cost per pig (FC/P) was determined by phase for each cluster. Then model calculated profit/loss with these values (Table 8). Cluster II had the highest FC/P in each phase; cluster I showed the lowest cost per pig in each phase; cluster III was in between.

Phase	n	Placement FC	Nursery FC	Early Finishing FC	Late Finishing FC	Total FC
**Cluster I**	6	$3.19	$3.28	$8.32	$12.59	$27.38
**Cluster II**	40	$4.69	$4.85	$12.95	$12.13	$34.62
**Cluster III**	14	$3.95	$4.10	$10.88	$13.69	$32.62

**Table 9 vetsci-06-00049-t009:** Modelled expenses and net profit per pig appear for each cluster. Total NDP is the average total number of dead pigs for each of the clusters.

	Total NDP	Total Pigs Marketed	Total Revenue per Pig Placed	Total Expense per Pig Placed	Total Profit per Pig Placed
**Cluster I**	304	696	$93.48	$106.53	–$13.05
**Cluster II**	56	914	$126.72	$115.18	$11.54
**Cluster III**	138	862	$117.84	$113.57	$4.27

**Table 10 vetsci-06-00049-t010:** Summary of management and performance data for each cluster. Average starting weight is the average pig weight at weaning upon placement into their WTF barn. Head placed is the average number of pigs placed per WTF barn, capacity place is a proportion based upon an expected number of head available to populate barn; m^2^/pig is the area per pig calculated from the pen dimensions in Table 1. Avg. total NDP is the average of all dead pigs for each cluster. Avg. market weight is the average sale weight for each pig within a cluster.

	n	Avg. Starting Weight, kg	Head Placed	Capacity Placed	m^2^/Pig	Avg. Total NDP	Average Cluster Mortality	Avg. Market Weight, kg
**Cluster I**	6	7.14 ^a^	2046 ^c^	1.11 ^e^	0.6197 ^g^	568	27.6 ^a^	110.22
**Cluster II**	40	6.61 ^b^	1864 ^d^	0.98 ^f^	0.6856 ^h^	152	8.1 ^b^	118.39
**Cluster III**	14	7.15 ^a^	2026 ^c^	1.06 ^e^	0.6271 ^g^	286	14.2 ^a^	116.12

^a,b,c,d,e,f,g,h^ Values within a column and with different superscripts are different (*p* < 0.05).

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
