# Peer review of "Mortality Patterns in a Commercial Wean-To Finish Swine Production System"

_vetsci, 2019, doi:10.3390/vetsci6020049_

Round 1
Reviewer 1 Report
Line 43-44: Neither this study was design to do that too. Your study was to describe and not to give a template on how to reduce mortality. By going with the objectives of the study that follow these line, I think you need to rewrite that section.
Author Response
We fully agree with your comment and have rewritten that section.
Reviewer 2 Report
This paper includes valuable results for understanding the mortality pattern in the commercial wean-to-finish swine production system. However, there are a few points that should be addressed prior to publication. Accordingly, I recommend publication of the manuscript after some revisions.
Major comments
1. The authors should explain more background about the meaningful of this study in the section of Introduction.
2. Why the stocking density in this study is lower than the recommended stocking densities for growing-finishing pigs in US. See reference: Cho, J. H., & Kim, I. H. (2011). Effect of stocking density on pig production. African Journal of Biotechnology, 10(63), 13688-13692.
Minor comments
1. L71, provide details of the housing of other 12 additional lots of pigs. Why and when?
2. L88-89 Each barn in site 7 contained……. Each barn in sites 1-6, and 8
3. L91 Figure 2?
4. L146-149 add some description about the results rather than point out where the results presented.
5. Table 3 and Table 5 N3, N4….should be defined
6. Table 2, 8, and 10: the headings in the table should be bold.
7. The description about the statistical analysis should be added, the results in Table 10.
8. L229-L232 Indeed, the high mortality is associated with the stocking density. However, for the cluster 1 and 3, the difference in stocking density is not that big, but the mortality number differed largely. Should discuss more about these observations. Otherwise, it can’t be concluded that stocking density contribute to the difference between cluster 2 and other two clusters.
Author Response
Thank you for your review.
See attached

Reviewer 3 Report
Line 76 Since these are Smithfield barns, there's a pretty good chance that they are partially slatted with shallow gutter, and that'd be important to mention since they are the only system that uses that type of design Line 97 Table 1 Title says Square Footage but correctly use m2 in the body of the table. Change square footage to Area Line 105 Diets switched based on days and not a feed budget. Is that an issue? Section 3.1 line 131 use weeks 13 and 17, but that doesn't match the groups. Maybe more text to clarify Table 5, etc In the Journal of Animal Science FCR is gain:feed, but here it's listed as feed:gain. Not sure which your journal requires Abstract says nothing about area/pig, but that's the main part of the Discussion section Were there any characterizations of reasons for death? That could be very helpful/insightfulAuthor Response
Thank you for your comments
